# Efficacy and moderators of mindfulness-based cognitive therapy (MBCT) in 'Difficult to Treat' depression: protocol for a systematic review and individual participant data meta-analysis of randomised controlled trials

Thorsten Barnhofer,[1] Maria Niemi [iD],[2] Johannes Michalak,[3] Maria Velana,[3] J Mark G Williams,[4] Alberto Chiesa,[5] Stuart Eisendrath,[6] Kevin Delucchi,[6] Zindel Segal,[7] Mira Cladder-Micus,[8] Anne Speckens [iD],[9] Ali Akbar Foroughi,[10] Mauro García-Toro,[11,12,13,14] Jesus Montero-Marin,[4,15,16] Barney Dunn [iD],[17] Clara Strauss,[18,19] Florian Alexander Ruths,[20] Mary Ryan,[21] Mathias Harrer[22,23,24]

Correspondence to
Dr Maria Niemi;
maria.niemi@ki.se

## ABSTRACT

**Introduction** About 30% of depressed patients suffer from a protracted course in which the disorder continues to cause significant burden despite treatment efforts. While originally developed for relapse prevention, mindfulness-based cognitive therapy (MBCT) has increasingly been investigated in depressed patients with such 'difficult-to-treat' courses. This is a protocol for an individual participant data (IPD) meta-analysis aiming to determine efficacy and potential moderators of MBCT treatment effects in this group based on evidence from randomised controlled trials.

**Methods and analysis** Systematic searches in PubMed, Web of Science, Scopus, PsycINFO, EMBASE and the Cochrane Controlled Trials Register for randomised controlled trials were completed on 17 June 2024. Authors of identified studies have contributed IPD, and data extractions have been completed. An update search will be conducted immediately before the start of data analyses. We will investigate the following outcomes: (a) self-reported and observer-reported severity of depression symptomatology, (b) remission and (c) clinically meaningful improvement and deterioration. One-stage and two-stage IPD-MA will be conducted with one-stage models using the observed IPD from all studies simultaneously as the primary approach. One-stage IPD models will include stratified study intercepts and error terms as well as random effects to capture between-study heterogeneity. Moderator analyses will test treatment-covariate interactions for both individual patient-level and study-level characteristics.

**Ethics and dissemination** The results will inform understanding of the use of MBCT in patients with current 'difficult-to-treat' depression and will contribute to arguments in favour of or against implementing MBCT as a treatment for this group. They will be published in a peer-reviewed journal and made available to stakeholders in accessible formats. No local ethical review was necessary following consultation with the Ethics and Governance Board of the University of Surrey. Guidance on patient data storage and management will be adhered to throughout.

**PROSPERO registration number** CRD42022332039.

## STRENGTHS AND LIMITATIONS OF THIS STUDY

⇒ The chosen approach using individual participant data has several advantages over conventional meta-analyses, including the possibility to assess whether individual participant-level characteristics influence treatment effects with more specificity and power.

⇒ Limitations of the approach include the fact that inconsistencies across studies regarding the format and type of variables reported may restrict our ability to investigate moderators of treatment effects.

⇒ Transformation of outcome measures to a common metric across all studies may come with a small loss of information.

## INTRODUCTION

Major depressive disorder (MDD) is highly prevalent, debilitating[1] and takes a recurrent or chronic course in about 30% of patients, in which established treatments fail to bring sustained remission.[2] Treatment resistance has traditionally been defined in narrow operational terms (eg, non-response after two adequate pharmacological treatments), but such definitions vary across studies and exclude many patients with persistent, impairing depression. The broader construct of difficult-to-treat depression (DTD) was recently introduced to capture the true

spectrum within a single heuristic, providing a clinically meaningful framework that better reflects the overlap between these groups and the challenges faced in practice.[3 4] Defined as 'depression that continues to cause significant burden despite usual treatment efforts', such courses are associated with continuing functional impairment, reduced quality of life and significantly increased risk of chronic physical and neurodegenerative disorders.[5 6] Finding effective treatments for DTD remains an important challenge for depression research.[7]

While treatment approaches for DTD have traditionally been dominated by pharmacological strategies,[8] it is now increasingly acknowledged that effective management requires a broader emphasis on working with symptoms, maximising function and minimising burden.[4] Psychological interventions have an important contribution to make in this context,[9] and emerging evidence suggests that mindfulness-based cognitive therapy (MBCT[10]), an 8-week treatment that combines elements of cognitive therapy with mindfulness meditation, may be suited for this purpose.[11] Using a mental training approach, the intervention aims to provide patients with portable skills that enable them to disengage from habitual maladaptive responses and respond more adaptively to negative mood and stress.

MBCT was originally introduced for the prevention of relapse in patients who are currently in remission and has been proven to be effective for this purpose,[12] particularly in patients with highly recurrent courses (a use that is endorsed by treatment guidelines internationally). Early trials restricted the intervention to patients who were currently in full remission, as it was assumed that intensive mindfulness practice may be too demanding for those who are suffering from current symptoms.[13 14] However, as patients who are at high risk of relapse are likely to show residual symptoms between episodes, this restriction was later loosened, and analyses across studies for relapse prevention show that treatment effects increase, rather than decrease, with level of residual symptoms.[12] From a theoretical perspective, this observation is in line with the view that mindfulness skills serve to buffer the effects of negative mood and stress,[15 16] which implies that effects should become more visible under conditions of higher symptoms or stress.

Based on this reasoning, research has extended the use of MBCT to patients with current depression described as 'difficult-to-treat'. Preliminary studies investigating MBCT in treatment-resistant and chronic depression were published in 2008[17] and 2009,[18] and several definitive randomised controlled trials (RCTs) have been conducted since then,[19–24] suggesting that evidence has accumulated to a point where the use of MBCT as a treatment for patients with current DTD can now be considered for guideline endorsement and wider implementation. However, conclusions from individual studies are constrained by heterogeneous definitions and settings, variable comparators and follow-up durations, inconsistent outcome reporting and limited power for

moderator analyses. It seems timely, therefore, to bring data from existing RCTs together and analyse outcome across studies.

For this purpose, we have pooled individual participant data (IPD) from these studies to conduct meta-analyses with one-stage random effects meta-analysis of IPD as our primary approach given its advantages over conventional aggregate data meta-analyses.[25–28] This IPD meta-analysis (IPD-MA) will apply a consistent DTD framework, harmonise outcomes, estimate overall effects using one-stage models and test prespecified patient-level moderators, with comprehensive risk-of-bias and small-study assessments. The main aim of the analyses is to establish efficacy of MBCT for patients with DTD as compared with the control conditions used in the existing trials. Given the need for both effective symptom reduction and the maintenance of gains, we will aim to determine immediate effects at the end of the 8-week MBCT intervention (post-treatment) and effects in the longer term after patients have finished the intervention and are free to continue engaging in mindfulness practice by themselves (follow-up).

An important advantage of IPD-MA is that it allows examination of moderator or subgroup questions across the pooled sample and can thus serve to overcome limitations in sample size and power for answering these questions in the original studies.[29] Information about potential moderators of treatment effects, such as severity of depression at entry to treatment or a history of bipolar disorder, is important as it can potentially help with implementation by providing more precise information about relative indications or contraindications of the intervention. A further aim of this IPD-MA is therefore to explore effects of potential individual patient-level (eg, sociodemographic and clinical characteristics) and study-level (eg, delivery format) moderators of treatment effects in the pooled dataset.

As the clinical concept of DTD does not link to an objective way of assessment, the current meta-analysis will include studies that have used a range of conventional definitions falling under the wider concept of DTD, including selecting participants based on treatment nonresponse/remission, treatment resistance or the presence of chronic depression. Subgroup analyses will allow us to determine the extent to which effects found in the aggregated data generalise across these conventional categories. Given that DTD reflects a wider construct encompassing a large and heterogeneous group of patients with different characteristics and different treatment histories, we will aim to carefully describe the sample both on the meta-analysis and the study levels.

## METHODS
### General approach
This IPD-MA is registered as part of a broader project on PROSPERO (registration number: CRD42022332039; date of registration in PROSPERO: 24 May 2022) and

any key changes or amendments will be documented there. In addition to the direct comparisons between MBCT and control conditions, the wider project also includes studies of an individual psychotherapeutic intervention—the cognitive–behavioural analysis system of psychotherapy—and considers a wider range of outcomes, including both beneficial and adverse treatment effects. The current protocol specifies procedures for the comparison between MBCT and the control conditions used in the existing MBCT trials only, as we assume that this comparison is of particular interest given the need for establishing further treatment options for DTD. The moderator and sensitivity analyses described here were prespecified in this protocol prior to data analysis but were not detailed in the initial PROSPERO entry. They are included to address theoretically and clinically relevant questions, and all analyses (including null results) will be reported to minimise selective reporting. The protocol is registered retrospectively as searches to identify eligible papers were already conducted in 2024, with the last search having been performed on 17 June 2024 and an update search planned immediately before the start of data analysis. Corresponding authors of the selected studies were contacted and asked to provide raw data from their studies. The Preferred Reporting Items for Systematic Reviews and Meta-Analyses IPD statement[30] will be followed for the reporting of this study.

## Eligibility criteria

### Types of studies to be included

This IPD-MA will include only peer-reviewed RCTs, published in English. There will be no restriction on the type of recruitment settings. The restriction to English language publications is due to feasibility constraints and is recognised as a potential source of language bias.

### Type of intervention

Included studies have investigated the effects of manualised MBCT alone or as an adjunct treatment to treatment as usual (TAU).[31] MBCT is a psychological group-based intervention that combines intensive training in mindfulness meditation with elements of cognitive–behavioural therapy (CBT) for depression. Mindfulness can be operationally defined as non-judgemental awareness of present moment experience.[32] As MBCT was originally designed to prevent relapse rather than to reduce symptoms during a current episode, we will accept studies that introduce minor adaptations to the *delivery* of the manualised approach (eg, incorporating specific support for managing current depressive symptoms, particularly around the engagement in regular practice, and rephrasing psychoeducational components to address current depression rather than relapse, *without altering the structure or core content of the manual*). Other types of mindfulness-based interventions will not be included. MBCT studies will be considered regardless of delivery format (eg, physical group setting vs videoconference).

### Types of comparators

This may include TAU for depression, including pharmacological and psychological treatments (eg, CBT) or active controls. The latter will include established interventions as well as control interventions designed to mimic effects of mindfulness-based interventions in some or all aspects, with the exception of the core element of mindfulness.

### Participants

We will include studies that recruited adults with current MDD[33] [34] and studies that included mixed groups of patients with residual depressive symptoms or full episodes in the context of a chronic or treatment-resistant course of the disorder. In the latter case, participants with only residual symptoms will be excluded from the primary analysis approach (one-stage meta-analysis). As DTD does not link to an objective way of assessment, we will opt for a maximally inclusive interpretation of the term and include any studies that selected patients based on previous treatment non-response/remission, treatment resistance or chronicity. We will exclude studies with participants below the age of 18 and studies that have exclusively recruited participants over the age of 65, given that the factors involved in the onset, maintenance and recurrence of depression in these age groups can differ from those in adults within this age bracket.[35] [36]

### Outcomes

The main type of outcome will be severity of depressive symptomatology as assessed using standard depression outcome measures. Standard depression outcome measures may include self-report measures and observer-rated measures. If both types of measures are available, we will investigate effects on both types of outcome measures. If several measures are available for a single study, preference will be given to the measure that was used as the primary outcome of the original study. To compare outcomes across studies using different outcome measures, we will transform scores from these measures into standardised scores using the common metric developed by Wahl *et al*.[37] Where a measure is not part of the common metric approach, we will transform the measure into one that is part of the common metric using published conversion tables,[38] and then derive the common metric.

To facilitate clinical interpretation of results, we will derive dichotomous outcomes reflecting remission, improvement and deterioration. To apply a common rule for all studies, common metric scores will be converted to the Patient Health Questionnaire 9 (PHQ-9) and dichotomous outcomes computed based on this measure. Following conventions for the PHQ-9 used in pragmatic trials and health services, remission will be defined as a score of less than 10,[39] [40] that is, a shift from 'caseness' to a non-case score. Definition of clinically meaningful change will follow suggestions from recent trials in treatment-resistant depression, which has indicated a

threshold of 6 or more points.[41 42] We will therefore define clinically meaningful improvement as a reduction of 6 or more points and clinically meaningful deterioration as an increase of 6 or more points on the PHQ-9.

A more detailed analysis of negative effects, including suicidality, is covered in a separate protocol covering a network meta-analysis of MBCT and another individual psychotherapeutic intervention (Michalak *et al*, in submission). Secondary outcomes such as mindfulness, self-compassion and quality of life will be collected and may be used for future analysis.

### Moderators

We will also investigate potential moderators of treatment response, that is, factors that determine whether a person responds better to MBCT as compared with the control condition or vice versa. These will include potential moderators on the individual patient level (eg, sociodemographic characteristics, history of childhood trauma and clinical characteristics) and on the study level. We will include patient-level characteristics in the analyses, if they are reported across a sufficient number of studies and datasets (see the Statistical analysis section for criteria).

Sociodemographic and clinical characteristics that we aim to investigate include age,[43] gender,[44] marital status,[45] education,[46] employment,[47] childhood adversity,[48] age at onset of depression,[49] number of previous episodes,[50] duration of current episode,[51] current suicidality,[52] current antidepressants,[53] past antidepressants and treatment resistance to antidepressants,[54] comorbid anxiety disorder, comorbid obsessive-compulsive disorder, comorbid post-traumatic stress disorder, comorbid substance abuse or dependence, comorbid eating disorder, comorbid personality disorder[43] and level of depressive symptoms at entry.[55] Study-level characteristics will include type of delivery (remote vs face to face), type of control condition (TAU, active control), type of DTD (non-response/remission, treatment resistance, chronic depression) and study quality.

### Timing of outcome assessments

As follow-up periods of the studies differ, we will group follow-up periods in the following categories: 8–10 weeks postrandomisation (post-treatment), 11–34 weeks postrandomisation (follow-up 1) and 35–60 weeks postrandomisation (follow-up 2).

### Searches and study selection

We searched PubMed, Web of Science, Scopus, PsycINFO, EMBASE and the Cochrane Controlled Trials Register using index and free terms, jointly with Boolean operators, on four tiers, namely: (1) depressive disorder, (2) non-response, treatment resistance and chronicity, (3) MBCT and (4) RCT. The search terms used for the different databases are shown in the online supplemental appendix. We did not search grey literature or trial registries beyond the Cochrane Controlled Trials Register, and we did not consult experts outside of the research team.

We searched for any study published until the day of the search (17 June 2024). There was no lower time limit. Initial screening of titles was performed by two independent reviewers (TB, JM), with the publication abstract being obtained if selected by at least one reviewer. Abstracts were screened by the same two independent reviewers, with disagreements about eligibility being referred to a third reviewer whose decision was final. An update search will be conducted immediately before the start of data analysis.

### Risk of bias assessment

The validity of the studies will be assessed using the Risk of Bias tool by the Cochrane Collaboration (version 2).[56 57] Possible sources of bias assessed by the tool are (1) random sequence generation, (2) allocation concealment, (3) blinding of outcome assessors, (4) incomplete outcome data, (5) selective outcome reporting and (6) other threats to validity, including lack of compliance and different timing of outcome assessments. Data from the published papers of the trials will be evaluated by two independent assessors. The agreement rate will be reported, and any disagreements will be solved through discussion. A sensitivity analysis will be performed, excluding studies judged to be at high risk of bias, or where risk of bias is unclear.

### Patient and public involvement

A lived experience advisory group, led by MR and consisting of experts who have been involved in previous trials of MBCT, will be consulted for the interpretation of study results and their dissemination. A coproduced lay summary of results will be written for dissemination alongside papers for peer-reviewed journals.

### IPD data collection and aggregation

All corresponding authors of the articles selected for inclusion were contacted via email by TB with an invitation to participate in the review and to share the IPD from their primary study. The standard letter described all relevant details of the study, including its purpose, main research questions and the variables of interest, and ended with a request to share the raw data from the study. Corresponding authors of studies were invited to serve as collaborators for the project. If corresponding authors did not respond within 4 weeks, we sent a second email request. If this had failed, a second author was contacted. Apart from one study, whose authors could not be contacted, none of the selected studies required contact attempts beyond a second author, and the responding authors agreed to share their data.

### Data checking and integrity

Authors were requested to provide a codebook or data dictionary explaining variable names, coding of values, measurement units, etc. After accepting the invitation to collaborate and signing the data transfer agreement, the study authors shared their IPD via secure data transfer portal at the University of Surrey. Data are stored securely

on University of Surrey servers in accordance with the European Union's General Data Protection Regulation Act and the UK's Data Protection Act 2018.

TB and MH will take primary responsibility for preparing and extracting outcome data and other characteristics, with MV independently checking extractions for accuracy. Received datasets will be reviewed to assess completeness and checked against the reported data from relevant study publications. The numbers of participants, overall and by arm, will be confirmed to correspond with the numbers in the study publication. Ranges of values will be inspected for clearly erroneous data. Baseline participant demographic data will be inspected by arm, with the aim of checking that the IPD matches the data reported in the publication, in terms of the numbers of participants with reported data, and overall aggregated values for each variable. Discrepancies will be referred to the study authors for investigation, and any remaining discrepancies will be recorded. For each included trial, we will extract information on funding sources and any reported conflicts of interest, and this information will be presented alongside other study characteristics.

Although the main search and initial receipt of IPD occurred before the submission of this protocol for publication, the protocol was registered in PROSPERO (CRD42022332039) beforehand. No data analysis or outcome-driven decisions were made prior to finalising the protocol, and the methods described here were prespecified prior to data inspection.

### Data harmonisation
Once datasets from the individual studies have been checked and outcomes standardised, they will be merged into the final dataset. The final dataset for analysis will be checked again for accuracy by a researcher in the study team.

### STATISTICAL ANALYSIS
To finalise the selection of moderator variables, we will determine the presence of these variables across the individual datasets and decide on ways to standardise these variables where needed, for example, by collapsing categories. Potential moderator variables will be included in the analyses if the variable has at least 40% available data in the final dataset and has been recorded by at least four studies.

To investigate the efficacy of MBCT compared with control conditions and test moderating effects of individual patient-level and study-level characteristics, we will conduct one-stage and two-stage IPD-MAs. Random effects models will be used throughout because we expect high statistical heterogeneity due to variation in control groups (TAU, active controls) and study populations (treatment non-responders/remitters, treatment resistance, chronic depression).

The main research question of the study (whether MBCT is efficacious compared with control conditions) will be addressed via a maximum of three pairwise comparisons: (1) MBCT versus TAU controls, (2) MBCT versus active controls and (3) MBCT versus all controls.

By default, all analyses will be conducted in a Bayesian framework using Just Another Gibbs Sampler/Bayesian analysis Using Gibbs Sampling.[58] We chose a Bayesian approach because it offers greater flexibility in the parametrisation of the (one-stage) IPD-MA models, and because this allows for a more intuitive interpretation of results based on posterior tail probabilities.[59] Across all analyses, findings will be considered 'significant' when the credibility interval of the relevant effect does not include zero.

### Two-stage random effects meta-analysis
We will first conduct a conventional meta-analysis using data extracted from the published papers. We will calculate Hedges' g to determine the effect sizes of the difference between the intervention and the control conditions reported in the papers. To investigate heterogeneity of effect sizes, we will calculate $I^2$, the percentage of variation not attributable to sampling error, as well as 95% prediction intervals around the pooled effect.[60 61] Leave-one-out analyses will be conducted to identify potential outliers and/or influential studies. Small-study effects will be investigated visually using funnel plot inspection and the Egger's test for asymmetry. A weakly informative Half-Cauchy (0, 0.5) prior will be used for the between-study heterogeneity variance $\tau^2$,[62] and flat normal priors for all other parameters. A detailed justification for this prior setup has been derived in previous work.[63]

### One-stage random effects meta-analysis using IPD
One-stage IPD-MA models will be used as the primary analytical approach. To analyse the effects on common metrics-converted symptom severity scores, an identity-link linear mixed effects model with stratified intercepts and trial-specific (heteroscedastic) error terms will be used. Let $y_{ik}$ be the depressive symptom severity of patient $i$ in trial $k$ (with $k=1, …, K$ studies included in the meta-analysis). The one-stage IPD can then be defined as:

$$y_{ik} = \alpha_k + \theta\, T_{ik} + u_k T_{ik} + \beta_k \left( X_{ik} - \bar{X}_{ik} \right) + \varepsilon_{ik} \qquad (1)$$

$$u_k \sim N\left(0, \tau_\theta^2\right) \quad \varepsilon_{ik} \sim N\left(0, \sigma_k^2\right)$$

where $T_{ik}$ is the treatment indicator, $X_{ik}$ is the symptom severity score at baseline, $\bar{X}_{ik}$ is the mean of $X$ for each specific trial, $\alpha_k$ is the stratified trial intercept and $u_k$ is the trial-specific random slope of the average treatment effect $\theta$. For this model, priors mirroring the two-stage approach will be used throughout, with a uniform distribution assigned to $\sigma_k^2$.

The same model specification, but with a logit link, will be used to model the effects on binary outcomes, including remission, clinically meaningful improvement and clinically meaningful deterioration based on PHQ-9 thresholds:

$$\log_e\left(\frac{\pi_{ik}}{1-\pi_{ik}}\right) = \begin{array}{l} \alpha_k + \theta T_{ik} + u_k T_{ik} \\ + \beta_k \left( X_{ik} - \overline{X}_{ik} \right) + \varepsilon_{ik} \end{array} \quad (2)$$

$$y_{ik} \sim \text{Bern}\left(\pi_{ik}\right) \quad u_k \sim N\left(0, \tau_\theta^2\right) \quad \varepsilon_{ik} \sim N\left(0, \sigma_k^2\right)$$

where $y_{ik}$ now represents the patient's binary outcome of interest. Numbers needed to treat corresponding to the estimated Standardised Mean Difference (SMD) values will be calculated using the method by Furukawa and Leucht,[64] and as the inverse of the absolute risk reduction otherwise.

## Moderator analyses

We will explore potential moderators of treatment effect by adding individual participant-level and study-level treatment-covariate interactions into the main IPD-MA model. To optimise power, these analyses will be done on treatment effects at post-treatment as all studies can provide data for this timepoint.

## Missing data

We will record the percentage of individual participant missing data for baseline characteristics and outcomes in each of the studies included in the meta-analyses. All analyses will be conducted according to the 'intention-to-treat' principle, so that analyses target a treatment policy estimand.[65] Missing data will be handled using multiple imputation (fully conditional specification; Multiple Imputation by Chained Equations (MICE) algorithm[66] under the missing at random assumption). For the main effectiveness analysis, multilevel two-stage imputation models with heteroscedastic errors will be used to account for the nested data structure.[67] Highly collinear variables will be removed as predictors, as well as variables with systematically missing information ('structural zeros'). A total of $m$=50 imputation sets will be generated. A separate imputation model will be constructed for each focal moderator, including auxiliary variables with a maximum of 10% missingness. Imputations will be generated separately for each of the treatment groups (intervention, control) to account for treatment-covariate interactions. All analysis models will be fitted in the multiply imputed data. Parameter estimates and test statistics will then be aggregated by mixing the draws from the posterior distribution of each completed dataset.[68] Sensitivity analyses will be performed to compare results using observed and imputed data.[69]

## Subgroup analyses

We plan to examine the efficacy of MBCT and potential moderators of treatment outcomes in subgroups defined by the conventional descriptive categories used by the individual studies to define their populations, distinguishing between studies that recruited patients characterised as treatment non-responders/remitters, treatment resistant or as suffering from chronic depression. These analyses will be based on subsamples created from the final IPD dataset.

## Sensitivity analyses

As a sensitivity analysis, we will repeat all main effectiveness analyses using equivalent frequentist (one and two-stage) approaches. Additionally, we will conduct sensitivity analyses to investigate whether sources of heterogeneity affect the overall effect size estimate and robustness of our findings. This will include exploring the role of risk of bias and other potential study-level sources of heterogeneity that may become obvious after data have been examined. These analyses will be conducted using a two-stage IPD-MA model.

## DISCUSSION

This IPD-MA will provide estimates of efficacy for the use of MBCT in DTD and offer comprehensive information about potential moderators of treatment effects. As such, it will contribute to arguments in favour of or against implementing MBCT as a treatment for DTD and may help decide whether MBCT is a recommendable treatment option for this group. This is important given the need to improve the care and management of DTD and the increasing recognition that psychological therapies can have an integral role in this endeavour. Information about potential moderators of treatment effects will help refine the understanding of who may or may not benefit from MBCT for DTD.

We have chosen the emerging concept of DTD as an umbrella because it provides a broad conceptualisation of depression that remains burdensome despite usual treatment attempts. Most available trials have operationalised this construct using narrower criteria such as non-response, non-remission, treatment resistance or chronicity, and our inclusion criteria therefore reflect how DTD has been studied to date. These characteristics represent core manifestations of DTD, and their use ensures consistency and feasibility for the planned meta-analysis. We acknowledge, however, that DTD extends beyond these dimensions, for example, in its emphasis on functional outcomes, and generalisability of findings to the broader construct should be interpreted with this in mind.

The chosen approach using IPD has several advantages over conventional meta-analyses, including the possibility to assess whether individual participant-level characteristics influence treatment effects with more specificity and power. Limitations of the approach include the fact that inconsistencies across studies regarding the format and type of variables reported may restrict our ability to investigate moderators of treatment effects and that transformation of outcome measures to a common metric across all studies may come with a small loss of information. To address this latter point, the current study will complement one-stage IPD analyses with a conventional two-stage approach. A more general concern that is often raised for IPD-MAs is inclusion bias, as it may be difficult to obtain IPD for all selected studies.

This IPD-MA is registered as part of a broader project that examines a wider range of interventions and outcomes, including both beneficial and adverse effects. The present protocol specifies procedures for the comparison between MBCT and control conditions in MBCT trials, as we assume that this comparison is of particular interest given the need to establish interventions for DTD. To our knowledge, this is the first IPD-MA to investigate the effects of MBCT in DTD and assess moderators of treatment effects. As evidence from trials of MBCT for DTD has now accumulated to a point where the use of MBCT for DTD can be considered for endorsement in treatment guidelines, findings from this IPD-MA have the potential to inform treatment recommendations and decisions regarding implementation. The results will provide clinicians, healthcare providers and people who are suffering from DTD with information on the characteristics that may make it more or less likely for patients to benefit from this treatment and thus facilitate implementation.

## ETHICS AND DISSEMINATION

No local ethical review was necessary following consultation with the Ethics and Governance Board of the University of Surrey. The investigators of the primary trials have local ethical approval for sharing the data. TB and MH will be responsible for oversight of the database and analyses will be conducted by MH and TB. Patient privacy will be ensured by adhering to University of Surrey guidance on data management and storage. Exchange of data is governed by interinstitutional data sharing agreements. All data were pseudonymised (deidentified) before sharing, shared through secure upload facilities and are stored on secure servers. The dataset will not be open access. Researchers of good standing may receive access pending institutional approvals on data transfer.

We will disseminate the work in peer-reviewed journals, provide lay summaries and discuss findings with patient and lived experience groups. Other means of communication to disseminate the results will include conference presentations and social media announcements.

Author affiliations
[1]University of Surrey, Guildford, UK
[2]Department of Global Public Health, Karolinska Institutet, Stockholm, Sweden
[3]Department of Psychology and Psychotherapy, Witten/Herdecke University, Witten, Germany
[4]Department of Psychiatry, University of Oxford, Oxford, UK
[5]Università di Bologna, Bologna, Italy
[6]University of California at San Francisco, San Francisco, California, USA
[7]Graduate Department of Psychological Clinical Science, University of Toronto–Scarborough, Toronto, Ontario, Canada
[8]Radboud University Nijmegen, Nijmegen, The Netherlands
[9]Psychiatry, Radboudumc Instituut voor Wetenschappelijk Onderwijs en Opleidingen, Nijmegen, The Netherlands
[10]Department of Clinical Psychology, Kermanshah University of Medical Sciences (KUMS), Kermanshah, Iran (the Islamic Republic of)
[11]Research Network on Chronicity, Primary Care and Health Promotion, Madrid, Spain
[12]University of the Balearic Islands, Palma, Spain
[13]Health Research Institute of the Balearic Islands, Balearic Islands, Spain
[14]Department of Medicine, University of the Balearic Islands, Palma, Spain
[15]Research & Innovation Unit, Parc Sanitari Sant Joan de Déu, Sant Boi de Llobregat, Spain
[16]Consortium for Biomedical Research in Epidemiology and Public Health, Madrid, Spain
[17]University of Exeter, Exeter, UK
[18]Sussex Partnership NHS Foundation Trust, Worthing, UK
[19]University of Sussex, Brighton, UK
[20]South London and Maudsley NHS Foundation Trust, London, UK
[21]Royal College of Psychiatrists, London, UK
[22]Department of Psychiatry and Psychotherapy, Technical University of Munich, Munich, Germany
[23]Department of Clinical, Neuro and Developmental Psychology, World Health Organization Collaborating Center for Research and Dissemination of Psychological Interventions, Geneva, Switzerland
[24]Vrije Universiteit, Amsterdam, The Netherlands

**Acknowledgements** JM-M has a 'Miguel Servet' research contract from the ISCIII (CP21/00080), and he is grateful to the Spanish CIBER of Epidemiology and Public Health (CBERESP CB22/02/00052<ISCIII) for their support.

**Contributors** TB, MN, JM, MV and MH conceptualised and designed the study. TB developed the search strategy and contacted the primary authors, who contributed to further refinement of the design and approach of the study. TB, MV and MH were responsible for building the database. MH provided statistical expertise and is responsible for data analyses. TB drafted the protocol manuscript with statistical input by MH, which was critically revised by all authors. All authors read and approved the final version. TB serves as the guarantor of the review.

**Funding** The authors have not declared a specific grant for this research from any funding agency in the public, commercial or not-for-profit sectors.

**Competing interests** TB is the coauthor of a book on mindfulness-based interventions and regularly offers related workshops. He is a coinvestigator on a programme grant evaluating an adapted mindfulness-based cognitive therapy (MBCT) course for adolescents with depression. MN authored a book chapter on mindfulness-based interventions and also offers workshops in this area. JM is the director of the Achtsamkeitsinstitut Ruhr (an institute offering mindfulness training) and principal investigator of several German Research Foundation (DFG) projects. He receives royalties from mindfulness books he has authored. JMGW codeveloped MBCT and was the founding director of the University of Oxford Mindfulness Centre. He receives royalties from books describing the MBCT programme and honoraria from lectures and training workshops on mindfulness. AC leads Istituto Mente Corpo, a professional practice that offers courses in mindfulness-based interventions. SE is the author of a book on MBCT for chronic and treatment-resistant depression. ZS receives book royalties from Guilford Press, workshop fees from the Centre for Mindfulness Studies and revenue from online sales at Mindful Noggin Inc, all related to his work as a co-founder of MBCT. AS is the founder and director of the Radboudumc Expertise Centre for Mindfulness and coinvestigator on several studies examining the effectiveness and implementation of mindfulness-based interventions for patients with psychological or somatic disorders, as well as healthcare professionals. JM-M is affiliated with the University of Oxford Mindfulness Research Centre. BD leads the University of Exeter AccEPT Clinic, which offers MBCT courses. CS is a member of a training organisation commissioned by NHS England to deliver MBCT training across NHS Talking Therapies services. CS coleads the Sussex Mindfulness Centre and is a coinvestigator on programme grants evaluating adapted MBCT courses for adolescents with depression and for NHS staff. FAR regularly provides workshops on mindfulness-based interventions. MH serves as a statistical consultant for HelloBetter/GetOn Institut für Gesundheitstrainings GmbH, a company implementing digital mental health therapeutics in routine care.

**Patient and public involvement** Patients and/or the public were not involved in the design, or conduct, or reporting, or dissemination plans of this research.

**Patient consent for publication** Not applicable.

**Provenance and peer review** Not commissioned; externally peer reviewed.

**ORCID iDs**
Maria Niemi https://orcid.org/0000-0001-5407-6981
Anne Speckens https://orcid.org/0000-0001-5266-1554
Barney Dunn https://orcid.org/0000-0002-0299-0920

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
