## [Reviewer comments · BMJ Open]

ARTICLE DETAILS

Title (Provisional)

Efficacy and Moderators of Mindfulness-Based Cognitive Therapy (MBCT) in 'Difficult to Treat' Depression: Protocol for a Systematic Review and Individual Patient Data Meta-Analysis of Randomised Controlled Trials

Authors

Barnhofer, Thorsten; Niemi, Maria; Michalak, Johannes; Velana, Maria; Williams, J Mark G; Chiesa, Alberto; Eisendrath, Stuart; Delucchi, Kevin; Segal, Zindel; Cladder-Micus, Mira; Speckens, Anne; Foroughi, Ali Akbar; García-Toro, M; Montero-Marin, Jesus; dunn, barney; Strauss, Clara; Ruths, Florian Alexander; Ryan, Mary; Harrer, Mathias

VERSION 1 - REVIEW

Reviewer	1
Name	Villalón López, Francisco
Affiliation	Diego Portales University
Date	17-Aug-2025
COI	FV is a certified instructor in Mindfulness-Based Cognitive Therapy (MBCT) by the Oxford Mindfulness Center. His professional activities include conducting workshops and lectures on mindfulness and compassion-based interventions.

“Efficacy and Moderators of Mindfulness-Based Cognitive Therapy (MBCT) in ‘Difficult to Treat’ Depression: Protocol for a Systematic Review and Individual Patient Data MetaAnalysis of Randomised Controlled Trials”

Registered PROSPERO: CRD42022332039

This is a systematic review and an individual patient data (IPD) meta-analysis of randomized controlled trials (RCTs) aimed at evaluating the efficacy and moderators of MBCT in difficult-to-treat depression (DTD). This is an interesting and relevant topic that may make valuable contributions to the field by identifying updated evidence. However critical issues are detected.

Introduction

The introduction is well written, clear, and concise, successfully identifying a research gap. However, some conceptual elements could be further developed to enhance clarity and avoid confusion.

The article focuses on difficult-to-treat depression (DTD); it would be helpful to clearly define this concept in the introduction. The authors cite two articles for this definition, published in 2020 and 2022. It is noteworthy that **bipolar disorder (BD)** is not carefully considered in this concept, despite the high prevalence of BD, the fact that an initial depressive episode is the most frequent presentation, and the worse outcomes associated with antidepressant use in this population. Consequently, some studies may reflect apparent improvements due to **antidepressant discontinuation**, rather than the intervention itself.

DTD is an emerging concept that refers to a **heterogeneous group of patients**, which includes those with treatment-resistant depression but encompasses a broader perspective on this population.

In paragraph 4, the use of the term “proven” could be softened, as DTD remains an emerging concept under active investigation.

Furthermore, some statements are not sufficiently evidence-based and may generate confusion. For example, in the second paragraph the article refers to DTD and suggests that

MBCT “may be particularly suited for this purpose,” yet references 8 and 9 are from 2013 and do not specifically support the suitability of MBCT for DTD.

To enrich the introduction and better highlight the evidence gap, it would be desirable to include the **most updated recommendations for DTD treatment**, while identifying their weaknesses and clarifying how this research may address the gap. For example:

<https://www.tandfonline.com/doi/full/10.1080/08039488.2021.1952303#abstract>.

The authors operationalized DTD as “on-treatment non-response/remission, treatment resistance, or the presence of chronic depression.” This requires further justification, since consensus definitions include a **broader set of factors**, such as number of failed treatments, comorbid medical and psychiatric conditions, adherence, variability of symptoms, family history, and functional impairment, among others.

Methods

The journal guidelines for protocol publications state that *“Protocol papers should report planned or ongoing studies. The dates of the study should be included in the manuscript.”*

However, the manuscript reports: *“The protocol is registered retrospectively as searches to identify eligible papers were already conducted in 2024, with the last search having been performed on 17 June 2024 and an update search planned immediately before the start of data analysis.”*

It may be beneficial to align the protocol more closely with PRISMA-P guidelines to strengthen transparency and methodological rigor.

This raises a **critical issue** that must be addressed by the editor. There are substantial discrepancies between the PROSPERO-registered protocol and the submitted manuscript.

- **Scope and interventions:** The PROSPERO title was “*An individual patient data (IPD) meta-analysis of Cognitive Behavioral Analysis System of Psychotherapy (CBASP) and Mindfulness-Based Cognitive Therapy (MBCT) for patients with chronic and treatment-resistant depression.*” In PROSPERO, CBASP was included, yet in the current manuscript it is omitted. This must be explicitly acknowledged and justified.
- **Definition of population:** DTD was not mentioned in the original PROSPERO protocol. The present manuscript introduces this concept without reference or justification. Clear criteria for DTD must be provided, given the broad and heterogeneous definitions in the literature and guideline recommendations.
- **Intervention and comparators:** In PROSPERO, CBASP was the only intervention defined, and MBCT was included as a comparator/control. In the current manuscript, MBCT is repositioned as the main intervention, while comparators differ from the original registration. This constitutes a major deviation.
- **Primary outcomes:** The original protocol listed as primary outcomes: self-reported or observer-rated clinical deterioration (e.g., PHQ-9, HDRS), suicidality, adverse events, and serious adverse events. These outcomes have been omitted in the current protocol without justification.
- **Secondary outcomes and moderators:** The registered protocol specified psychological moderators (sex and pretreatment depression severity) and predefined sensitivity analyses (e.g., trauma history). The submitted protocol expands outcomes and introduces new moderators post hoc, increasing the risk of selective reporting. If a

DTD approach is adopted, it would be necessary and parsimonious to include variables recommended in DTD guidelines, such as functional outcomes, which are currently missing.

- **Methodological clarity:**

- Language restrictions are applied but not justified, contrary to AMSTAR 2 standards.
- The term “*minor changes to the manual*” for MBCT is unclear; more systematic detail is required for reproducibility.
- Search and study selection methods lack transparency: it is unclear how disagreements between reviewers were resolved, whether gray literature or trial registries were searched, or if expert consultation was used.
- Data extraction procedures are insufficiently described, and it is unclear if two independent extractors were used.
- Risk of bias assessment is mentioned, but it is unclear whether results will be incorporated into sensitivity analyses. Similarly, plans for assessing publication bias are unclear.
- Reporting of funding sources and conflicts of interest of included trials is absent. Given the potential for *allegiance bias*, this information is essential.

Discussion

It seems that the DTD approach represents a broader construct, whereas this study protocol addresses primarily treatment resistance, non-remission, and chronic depression. This discrepancy may be confusing to readers and requires clearer justification. The decision

to frame the study under the DTD concept should be explicitly explained, and additional variables relevant to DTD (e.g., functional outcomes) should be considered to adequately capture this complex clinical entity. Furthermore, critical deviations from the pre-registered protocol have been detected and must be transparently acknowledged and justified.

VERSION 1 - AUTHOR RESPONSE

Reviewer: 1

Dr. Francisco Villalón López, Diego Portales University

Comments to the Author:

Peer review

“Efficacy and Moderators of Mindfulness-Based Cognitive Therapy (MBCT) in ‘Difficult to Treat’ Depression: Protocol for a Systematic Review and Individual Patient Data MetaAnalysis of Randomised Controlled Trials”

Registered PROSPERO: CRD42022332039

This is a systematic review and an individual patient data (IPD) meta-analysis of randomized controlled trials (RCTs) aimed at evaluating the efficacy and moderators of MBCT in difficult-to-treat depression (DTD). This is an interesting and relevant topic that may make valuable contributions to the field by identifying updated evidence. However critical issues are detected.

Thank you for your positive remarks about the topic. We were glad to hear that the planned research may make a valuable contribution to the field. We are grateful for your careful review and appreciate the constructive comments that you have provided.

Introduction

The introduction is well written, clear, and concise, successfully identifying a research gap. However, some conceptual elements could be further developed to enhance clarity and avoid confusion.

The article focuses on difficult-to-treat depression (DTD); it would be helpful to clearly define this concept in the introduction. The authors cite two articles for this definition, published in 2020 and 2022. It is noteworthy that bipolar disorder (BD) is not carefully considered in this concept, despite the high prevalence of BD, the fact that an initial depressive episode is the most frequent presentation, and the worse outcomes associated with antidepressant use in this population. Consequently, some studies may reflect apparent improvements due to antidepressant discontinuation, rather than the intervention itself. DTD is an emerging concept that refers to a heterogeneous group of patients, which includes those with treatment-resistant depression but encompasses a broader perspective on this population.

Thank you for pointing us towards the need to define DTD in more detail. We follow the definition provided in the international consensus statement by McAllister-Williams et al. (2020), which describes DTD as “depression that continues to cause significant burden

despite usual treatment efforts". This definition focusses explicitly on MDD, although the statement acknowledges that a similar rationale may apply to bipolar disorder. As suggested, we have included this definition in the text, where it now appears in the first paragraph of the introduction (p. 7, para. 1).

"Major Depressive Disorder (MDD) is highly prevalent, debilitating [1] , and in about 30% of patients takes a recurrent or chronic course in which established treatments fail to bring sustained remission [2] . Recently framed under the broader heuristic of difficult-to-treat depression (DTD) [3, 4] , defined as "depression that continues to cause significant burden despite usual treatment efforts", such courses are associated with continuing functional impairment, reduced quality of life, and significantly increased risk for chronic physical and neurodegenerative disorders [5, 6] ."

We also appreciate the insightful comments on potential influences of bipolar disorder. Rather than adding a discussion of potential influences of bipolar disorder, we have decided to address this issue but making it more transparent how issues like this are addressed empirically in the meta-analysis. The IPD meta-analysis will include diagnosis of bipolar disorder/bipolar history as a potential moderator. We specifically chose the IPD approach because it allows such issues to be addressed empirically and we are confident that the proposed methods will provide a way of dealing with this concern. To make readers aware of this, we have added "history of bipolar disorder" to the examples given in the paragraph that introduces moderator analyses (p. 8, para. 4):

"An important advantage of IPD meta-analysis is that it allows to examine moderator or subgroup questions across the pooled sample and can thus serve to overcome limitations in sample size and power for answering these questions in the original studies [29] . Information about potential moderators of treatment effects, such as severity of depression at entry to treatment or a history of bipolar disorder, is important as it can potentially help with implementation by providing more precise information about relative indications or contraindications of the intervention."

Please also note that 4 of the 8 studies identified through the searches used bipolar disorder as an exclusion criterion.

In paragraph 4, the use of the term "proven" could be softened, as DTD remains an emerging concept under active investigation.

Thank you for highlighting the potential for ambiguity here. Our use of "proven" in this sentence was not intended as a strong epistemic claim rather in the everyday sense of "shown to be". To avoid confusion, we have rephrased the sentence as follows (p. 8, para. 2):

"Based on this reasoning, research has extended the use of MBCT to patients with current depression described as 'difficult-to-treat.'"

Furthermore, some statements are not sufficiently evidence-based and may generate confusion. For example, in the second paragraph the article refers to DTD and suggests that MBCT "may be particularly suited for this purpose," yet references 8 and 9 are from 2013 and do not specifically support the suitability of MBCT for DTD.

Thank you for this helpful observation. The sentence in question was intended to introduce the assumption that MBCT is a promising candidate as a further-line treatment in DTD rather than to pre-empt the results of our meta-analysis. To avoid it being read as a strong claim, we have softened the wording by replacing “it has been suggested” with “emerging evidence suggests”, and we now cite the largest study of MBCT for DTD to date to link the statement with evidence. We have also removed the term “particularly”, which further softens the statement. The revised sentence reads as follows (p. 7, para. 2):

“Psychological interventions have an important contribution to make in this context [8] and **emerging evidence suggests** that Mindfulness-Based Cognitive Therapy (MBCT [9]), an eight-week treatment that combines elements of cognitive therapy with mindfulness meditation, may be suited for this purpose [10] .

The study now cited in this context is:

10. Barnhofer T, Dunn BD, Strauss C, Ruths FA, Barrett B, Ryan M, et al. Mindfulness-based cognitive therapy versus treatment as usual after non-remission with NHS Talking Therapies high-intensity psychological therapy for depression: a UK-based clinical effectiveness and cost-effectiveness randomised, controlled, superiority trial. *Lancet Psychiatry*. 2025;12:433–46. [https://doi.org/10.1016/s2215-0366\(25\)00105-1](https://doi.org/10.1016/s2215-0366(25)00105-1).

To enrich the introduction and better highlight the evidence gap, it would be desirable to include the most updated recommendations for DTD treatment, while identifying their weaknesses and clarifying how this research may address the gap. For example: <https://www.tandfonline.com/doi/full/10.1080/08039488.2021.1952303#abstract>.

Thank you for directing us to this review. We now cite this work in the introductory paragraph to better highlight the evidence gap (p. 7, para. 1):

“Major Depressive Disorder (MDD) is highly prevalent, debilitating [1] , and in about 30% of patients takes a recurrent or chronic course in which established treatments fail to bring sustained remission [2] . Recently framed under the broader heuristic of difficult-to-treat depression (DTD) [3, 4] , defined as “depression that continues to cause significant burden despite usual treatment efforts” such courses are associated with continuing functional impairment, reduced quality of life, and significantly increased risk for chronic physical and neurodegenerative disorders [5, 6] . **Finding effective treatments for DTD remains an important challenge for depression research [7] .”**

The authors operationalized DTD as “treatment non-response/remission, treatment resistance, or the presence of chronic depression.” This requires further justification, since consensus definitions include a broader set of factors, such as number of failed treatments, comorbid medical and psychiatric conditions, adherence, variability of symptoms, family history, and functional impairment, among others.

Thank you for raising this important issue, which prompted us to further clarify our operationalisation of DTD. The consensus statement notes that “*this broad umbrella definition of DTD encompasses a large and heterogeneous group of patients, with different characteristics, different treatment histories, and different levels of ongoing burden.*” While it

highlights that “numerous” patient and illness characteristics may be associated with DTD, it does not specify which must be present for depression to be classified as difficult-to-treat. In other words, there is no *formal* requirement for a particular set of additional characteristics to be met. For this reason, we have operationalised DTD on the basis of the conventional definitions it subsumes.

We agree that it is nevertheless important to capture other characteristics frequently associated with DTD. We have therefore made it explicit in the text that information on illness and treatment histories will be reported at both the meta- and the study-level. Specifically, we now state at the end of the introduction (p. 9, para. 2):

“Given that DTD reflects a wider construct encompassing a large and heterogeneous group of patients with different characteristics and different treatment histories, we will aim to carefully describe the sample both on the meta- and the study-level.”

Further additions to the discussion section explain how the broader concept of DTD relates to the narrower conventional definitions used as selection criteria in our meta-analysis (see further below).

Methods

The journal guidelines for protocol publications state that “Protocol papers should report planned or ongoing studies. The dates of the study should be included in the manuscript.” However, the manuscript reports: “The protocol is registered retrospectively as searches to identify eligible papers were already conducted in 2024, with the last search having been performed on 17 June 2024 and an update search planned immediately before the start of data analysis.”

We would like to clarify that at the point of submission to *BMJ Open*, shortly after the current version of the protocol was uploaded to MedRxiv on 9 June 2025, the study was ongoing. Moreover, the wider project had already been registered on PROSPERO prior to the literature searches.

We agree that it would have been preferable to publish the protocol before commencing searches. To ensure transparency, we have explicitly stated that the protocol was registered retrospectively. While not ideal, this approach is consistent with other IPD meta-analysis protocols published in *BMJ Open* (e.g., Breedvelt JJF, Warren FC, Brouwer ME, et al. *BMJ Open*2020;10:e034158. doi:10.1136/bmjopen-2019-034158).

It may be beneficial to align the protocol more closely with PRISMA-P guidelines to strengthen transparency and methodological rigor.

We fully agree with the importance of aligning with PRISMA-P guidelines, and we submitted a completed PRISMA-P checklist with the manuscript. We are happy to make further adjustments to strengthen transparency and methodological rigor, and would be grateful for the indication of specific areas where additional clarification is required.

This raises a critical issue that must be addressed by the editor. There are substantial discrepancies between the PROSPERO-registered protocol and the submitted manuscript.

• Scope and interventions: The PROSPERO title was “An individual patient data (IPD) meta-analysis of Cognitive Behavioral Analysis System of Psychotherapy (CBASP) and Mindfulness-Based Cognitive Therapy (MBCT) for patients with chronic and treatment-resistant depression.” In PROSPERO, CBASP was included, yet in the current manuscript it is omitted. This must be explicitly acknowledged and justified.

We agree that the PROSPERO registration covers a broader project that includes both MBCT and CBASP, whereas the present manuscript focuses only on MBCT. The PROSPERO entry was designed to capture the full scope of the project, which includes comparisons between MBCT, CBASP, and their respective control conditions, across both positive and negative outcomes. It is common for large, comprehensive projects of this kind to report results for specific comparisons separately. In line with this approach, the present protocol addresses only MBCT versus control conditions, as this comparison is of particular clinical and scientific interest. We have now made this distinction explicit in the manuscript (p. 9, para. 3):

“This IPD meta-analysis is registered **as part of a broader project** on PROSPERO (registration number: CRD42022332039; date of registration in PROSPERO: 24 May 2022) and any key changes or amendments will be documented there. **In addition to the direct comparisons between MBCT and control conditions, the wider project also includes studies of an individual psychotherapeutic intervention, Cognitive Behavioral Analysis System of Psychotherapy, and considers a wider range of outcomes, including both beneficial and adverse treatment effects. The current protocol specifies procedures for the comparison between MBCT and the control conditions utilized in the existing MBCT trials only as we assume that this comparison is of particular interest given the need for establishing further treatment options for DTD.**”

• Definition of population: DTD was not mentioned in the original PROSPERO protocol. The present manuscript introduces this concept without reference or justification. Clear criteria for DTD must be provided, given the broad and heterogeneous definitions in the literature and guideline recommendations.

We acknowledge that the original PROSPERO registration did not use the term “difficult-to-treat depression” (DTD). At that time, the registration relied on conventional definitions that fall under the DTD umbrella. Since then, the DTD framework has gained wider acceptance and provides a more precise and clinically meaningful perspective than alternative terms such as “treatment resistance.” We therefore use DTD as an umbrella term in the present manuscript, while keeping the underlying inclusion criteria identical to those in the PROSPERO entry. As discussed further above, we now provide an explicit definition of DTD early in the manuscript and clarify how we will report study and patient characteristics commonly associated with this construct (see our responses to the comments about the definition of DTD further above).

• Intervention and comparators: In PROSPERO, CBASP was the only intervention defined, and MBCT was included as a comparator/control. In the current manuscript, MBCT is

repositioned as the main intervention, while comparators differ from the original registration. This constitutes a major deviation.

We acknowledge the difference in how interventions are presented between the PROSPERO registration and the current protocol. The PROSPERO record describes a broader project that includes a network meta-analysis of MBCT, CBASP, and their respective controls. Within that framework, MBCT was listed as a comparator to CBASP.

The present protocol focuses on a specific subset of that project: trials directly comparing MBCT with control conditions. In this context, MBCT is the primary intervention of interest, with various control conditions serving as comparators. This represents a refinement in scope rather than a change in the underlying research plan, and selection of studies and all analyses remain consistent with the broader PROSPERO registration.

• Primary outcomes: The original protocol listed as primary outcomes: self-reported or observer-rated clinical deterioration (e.g., PHQ-9, HDRS), suicidality, adverse events, and serious adverse events. These outcomes have been omitted in the current protocol without justification.

We acknowledge that the outcomes of clinical deterioration, suicidality, adverse events, and serious adverse events were listed as primary outcomes in the PROSPERO registration but are not included in the present protocol. This is because a detailed analysis of negative effects across MBCT, CBASP, and other control conditions is planned as a separate component of the wider project.

The procedures for this analysis have been prespecified in a separate protocol (<https://www.medrxiv.org/content/10.1101/2025.06.09.25329260v1>). These outcomes have therefore not been omitted, but will be reported in full and transparently in that dedicated analysis. All separate protocols will be linked to the PROSPERO registration.

• Secondary outcomes and moderators: The registered protocol specified psychological moderators (sex and pretreatment depression severity) and predefined sensitivity analyses (e.g., trauma history). The submitted protocol expands outcomes and introduces new moderators post hoc, increasing the risk of selective reporting. If a DTD approach is adopted, it would be necessary and parsimonious to include variables recommended in DTD guidelines, such as functional outcomes, which are currently missing.

We agree with the reviewer that there are differences between our PROSPERO registration and the submitted protocol. This reflects the fact that the PROSPERO record was registered as part of a broader project, whereas the current protocol focuses on a particular aspect of that project. The additional moderator analyses were prespecified in the protocol prior to data analysis and are motivated by theoretical considerations and prior empirical evidence, rather than arising post hoc from the data. Note that each of the moderators is introduced with references to previous research that has provided evidence for the variable to be related to treatment outcome in depression. We will report all analyses transparently, including non-significant findings, to mitigate the risk of selective reporting. To make this explicit, we have added the following sentences to the manuscript (p. 10, para. 1):

“The moderator and sensitivity analyses described here were prespecified in this protocol prior to data analysis but were not detailed in the initial PROSPERO entry. They are included to address theoretically and clinically relevant questions, and all analyses (including null results) will be reported to minimise selective reporting.”

• *Methodological clarity:*

o Language restrictions are applied but not justified, contrary to AMSTAR 2 standards.

We thank the reviewer for highlighting this. We restricted our search to articles published in English. This decision was made for feasibility reasons, as our team did not have the resources to translate and reliably extract individual patient data from non-English publications. We acknowledge that this may introduce a risk of language bias and have now made this limitation explicit in the manuscript, in line with AMSTAR 2 recommendations (p. 10, para. 2):

“This IPD meta-analysis will include only peer-reviewed RCTs, published in English. There will be no restriction on the type of recruitment settings. **The restriction to English-language publications is due to feasibility constraints and is recognised as a potential source of language bias.**”

o The term “minor changes to the manual” for MBCT is unclear; more systematic detail is required for reproducibility.

Minor adaptations of MBCT are necessary when the programme is delivered to patients with current depression, compared to its original use in those in remission. In such cases, facilitators need to take account of the impact of current symptoms when supporting patients’ practice, and psychoeducational material should be framed around managing present difficulties rather than relapse prevention. These modifications typically involve rewording rather than substantive changes. Importantly, the overall structure and core content of MBCT remain unchanged, ensuring that delivery still follows the original manual. We have clarified this in the manuscript (p. 11, para. 1) by adding:

“As MBCT **was** originally designed to prevent relapse **rather than** to reduce symptoms during a **current** episode, we will accept studies that introduce minor **adaptations** to the **delivery of the manualised approach** (e.g. incorporating specific support for managing current depressive symptoms, **particularly around engagement in regular practice, and rephrasing psychoeducational components to address current depression rather than relapse, without altering the structure or core content of the manual**). **Other** types of mindfulness-based interventions **will not be included**. MBCT studies **will be considered** regardless of delivery format (e.g. physical group setting versus videoconference).”

o Search and study selection methods lack transparency: it is unclear how disagreements between reviewers were resolved, whether gray literature or trial registries were searched, or if expert consultation was used.

We did not search grey literature or trial registries beyond the Cochrane Controlled Trials Register, and we did not consult experts outside of the research team, which itself includes a large number of experts in the field. This information has now been added to the section

describing searches and selection (p. 14, para. 1). Disagreements between reviewers regarding eligibility were resolved by a third reviewer, whose decision was final, which is stated on p. 14 in para. 2.

“We searched PubMed, Web of Science, Scopus, PsychINFO, EMBASE, and the Cochrane Controlled Trials Register using index and free terms, jointly with Boolean operators, on four tiers, namely: (1) depressive disorder, (2) non-response, treatment resistance and chronicity, (3) mindfulness-based cognitive therapy, and (4) randomised-controlled trial. The search terms used for the different databases are shown in the appendix. **We did not search grey literature or trial registries beyond the Cochrane Controlled Trials Register, and we did not consult experts outside of the research team.**”

o Data extraction procedures are insufficiently described, and it is unclear if two independent extractors were used.

Thank you for making us aware of this omission. This information has now been added to the section on data checking and integrity (p. 15, para. 4):

“TB and MH will take primary responsibility for preparing and extracting outcome data and other characteristics, with MV independently checking extractions for accuracy.”

Please note that we use the future tense here to reflect the status at the time of initial submission of the manuscript.

o Risk of bias assessment is mentioned, but it is unclear whether results will be incorporated into sensitivity analyses. Similarly, plans for assessing publication bias are unclear.

We would like to clarify that this information is already included in the manuscript. At the end of the section on risk of bias assessment (p. 14, para. 3), we state: “A sensitivity analysis will be performed excluding studies judged to be at high risk of bias, or where risk of bias is unclear.”

Similarly, our approach to publication bias is described in the section on two-stage random effects meta-analysis (p. 17, para. 3): “Small-study effects will be investigated visually using funnel plot inspection and the Egger’s test for asymmetry.”

o Reporting of funding sources and conflicts of interest of included trials is absent. Given the potential for allegiance bias, this information is essential.

Thank you for this comment. We note that the lead authors of the included trials are co-authors on this protocol and have declared their competing interests in the “Competing Interests Statement” at the end of the manuscript. To address the concern more directly, we have now clarified in the protocol that we will extract and report fundings sources and conflicts of interest for each included trial, so that potential influences such as allegiance bias can be considered in the interpretation of results.

The following sentence has been added (p. 16, para. 1):

“For each included trial, we will extract information on funding sources and any reported conflicts of interest, and this information will be presented alongside other study characteristics.”

Discussion

It seems that the DTD approach represents a broader construct, whereas this study protocol addresses primarily treatment resistance, non-remission, and chronic depression. This discrepancy may be confusing to readers and requires clearer justification. The decision to frame the study under the DTD concept should be explicitly explained, and additional variables relevant to DTD (e.g., functional outcomes) should be considered to adequately capture this complex clinical entity.

Thank you, as requested, we have added a paragraph that explains the relation between our criteria and the broader construct of DTD, including its implications for interpretation of findings. The paragraph (p. 20, para. 1) reads as follows:

“We have chosen the emerging concept of DTD as an umbrella because it provides a broad conceptualisation of depression that remains burdensome despite usual treatment attempts. Most available trials have operationalised this construct using narrower criteria such as non-response, non-remission, treatment-resistance, or chronicity, and our inclusion criteria therefore reflect how DTD has been studied to date. These characteristics represent core manifestations of DTD, and their use ensures consistency and feasibility for the planned meta-analysis. We acknowledge, however, that DTD extends beyond these dimensions, for example in its emphasis on functional outcomes, and generalisability of findings to the broader construct should be interpreted with this in mind.”

Furthermore, critical deviations from the pre-registered protocol have been detected and must be transparently acknowledged and justified.

As discussed further above, the analyses described in the current protocol are part of a broader project registered on PROSPERO. In addition to the information added in the Methods section, we have added the following sentences to the discussion (p. 21, para. 3):

“This IPD meta-analysis is registered as part of a broader project that examines a wider range of interventions and outcomes, including both beneficial and adverse effects. The present protocol specifies procedures for the comparison between MBCT and control conditions in MBCT trials, as we assume that this comparison is of particular interest given the need to establish interventions for DTD. To our knowledge, this is the first IPD meta-analysis to investigate the effects of MBCT in DTD and assess moderators of treatment effects.”

VERSION 2 - REVIEW

Reviewer	1
Name	Villalón López, Francisco

Affiliation **Diego Portales University**

Date **19-Sep-2025**

COI

Efficacy and Moderators of Mindfulness-Based Cognitive Therapy (MBCT) in ‘Difficult-to-Treat’ Depression: Protocol for a Systematic Review and Individual Patient Data Meta-analysis of Randomised Controlled Trials” **Registered PROSPERO:** CRD42022332039

Thank you for the responses and revisions. The manuscript has improved by incorporating several of the recommendations, including: refining the definition of DTD, softening terminology, adding references linking MBCT and DTD, inclusion of PRISMA-P, justification of language restrictions, acknowledgment of AMSTAR-2 limitations, clarification of minor adaptations to the MBCT manual, and specification of data extraction responsibilities.

However, there remain unresolved issues—3 minor and 2 critical.

1. **Minor:**

- The search should be updated, as more than one year has elapsed since the last search.
- **Justification and research gap:** The introduction conflates treatment resistance with the broader DTD construct, risking conceptual ambiguity. Without careful justification, this could lead to misinterpretation of findings and overgeneralization. The rationale for focusing on DTD remains underdeveloped. The added paragraph does not sufficiently explain how this protocol addresses weaknesses in existing evidence or fills a specific gap. The justification must be made explicit: why MBCT in DTD, why now, and how this work will advance the field beyond prior reviews.
- “Minor adaptations” of MBCT for current episodes need operational criteria for reproducibility (what qualifies as “minor”? who adjudicates? thresholds for exclusion). Without this, between-study heterogeneity risks becoming uninterpretable.

2. **Critical:** There are major discrepancies between the PROSPERO registration and the current protocol:

- **Population:** The introduction of the DTD construct represents a substantial shift from the original registration, which did not specify this population.
- **Primary outcomes:** The selective reporting of outcomes—particularly the omission of deterioration, suicidality, and adverse events, which are relegated to a separate protocol—constitutes outcome switching.

- **Interventions and comparators:** MBCT is repositioned from comparator to primary intervention, while CBASP, originally central in PROSPERO, is omitted here.
- **Moderators:** Additional moderators not prespecified in PROSPERO are introduced, raising concerns of post hoc rationalization.
- **Allegiance and COI:** Multiple investigators have deep MBCT involvement. The plan to extract trial-level funding/COI is welcome but must include **a priori sensitivity analyses** (e.g., excluding high-allegiance studies; meta-regression on allegiance).

Although the authors frame these as “refinements” within a broader project, splitting outcomes and interventions across multiple protocols risks diluting the integrity of prospective registration. The protocol, in its current form, may no longer function as a prospective safeguard against bias.

Importantly, BMJ Open guidelines (2025) state that “*Protocol papers should report planned or ongoing studies. The dates of the study should be included in the manuscript.*” The retrospective registration after searches had already been conducted, combined with the deviations outlined above, is inconsistent with this principle.

Recommendation: Major revision. The protocol and the PROSPERO entry must be fully aligned—either by updating PROSPERO to reflect the actual scope, population, outcomes, and analyses of this protocol, or by revising the manuscript to correspond faithfully to the original registration. Separate protocols for different interventions or outcomes should not be used as a workaround. Furthermore, the manuscript should clearly articulate the research gap in the treatment of DTD, acknowledging the weaknesses of current evidence and specifying how this review will address them. Full alignment and a stronger rationale are necessary to preserve the integrity and purpose of prospective registration.

VERSION 2 - AUTHOR RESPONSE

Reviewer: 1

Dr. Francisco Villalón López, Diego Portales University

Comments to the Author:

Efficacy and Moderators of Mindfulness-Based Cognitive Therapy (MBCT) in ‘Difficult-to-Treat’ Depression: Protocol for a Systematic Review and Individual Patient Data Meta-analysis of Randomised Controlled Trials” Registered PROSPERO: CRD42022332039

Thank you for the responses and revisions. The manuscript has improved by incorporating several of the recommendations, including: refining the definition of DTD, softening terminology, adding references linking MBCT and DTD, inclusion of PRISMA-P, justification of language restrictions, acknowledgment of AMSTAR-2 limitations, clarification of minor adaptations to the MBCT manual, and specification of data extraction responsibilities.

However, there remain unresolved issues—3 minor and 2 critical.

1. Minor:

o The search should be updated, as more than one year has elapsed since the last search.

This is already addressed in the protocol. We state in the abstract, in the “General approach” section of the “Methods”, and in the “Searches and study selection” section, that an update search will be conducted immediately before the start of data analysis.

o Justification and research gap: The introduction conflates treatment resistance with the broader DTD construct, risking conceptual ambiguity. Without careful justification, this could lead to misinterpretation of findings and overgeneralization. The rationale for focusing on DTD remains underdeveloped. The added paragraph does not sufficiently explain how this protocol addresses weaknesses in existing evidence or fills a specific gap. The justification must be made explicit: why MBCT in DTD, why now, and how this work will advance the field beyond prior reviews.

We respectfully note that our scope has consistently included both chronic **and** treatment-resistant depression, as reflected in the title and objectives of the original PROSPERO registration (24 May 2022).

This wider scope was chosen because evidence shows substantial overlap between these categories (for recent evidence see for example the studies by Lundberg et al., 2022, and Rose et al., 2024). Furthermore, treatment resistance was intended in its broadest sense as reliance on a single narrow definition of treatment resistance, such as two or more failed pharmacological treatments, would risk excluding large groups of patients who present with persistent, impairing depression but do not neatly fit such definitions, thereby limiting generalisability.

The construct of DTD, “depression that continues to cause significant burden despite usual treatment efforts”, was introduced to address precisely these problems of semantic and operational ambiguity. As stated in the protocol, DTD is best understood as a heuristic. It provides a more flexible, multidimensional and longitudinal definition in line with our initial intention behind the use of combined terms. Since the initial registration of our project in 2022, DTD has become the preferred term in the field, and we have therefore aligned our terminology in the protocol with this change.

Importantly, this represents a change in **terminology** rather than in the scope of our inclusion criteria, which remain defined in terms of conventional categories of chronic depression and treatment resistance in its broadest form (including treatment non-response and non-remission after one treatment, consistent with the Maudsley definition of treatment resistance). We confirmed this operationalisation with one of the authors of the consensus statement on DTD, Prof Allan Young, who has endorsed its plausibility and appropriateness.

We have added the following information to the introduction to clarify this wider context (p. 7, para. 1):

“Treatment resistance has traditionally been defined in narrow operational terms (e.g. non-response after two adequate pharmacological treatments), but such definitions vary across studies and exclude many patients with persistent, impairing depression. The broader construct of difficult-to-treat depression (DTD) was recently introduced to capture the true spectrum within a single heuristic,

providing a clinically meaningful framework that better reflects the overlap between these groups and the challenges faced in practice.”

With regard to the “*why MBCT in DTD, why now, and how this work will advance the field beyond prior reviews*”, the introduction clearly outlines that there is a lack of established psychological treatment options for DTD, that there is now a critical mass of studies (8 RCTs, including two large recently published trials) investigating MBCT in patients with DTD, that it is therefore timely to synthesise evidence, and that the synthesised evidence is likely to be sufficient to inform decisions about treatment guideline endorsement, which can potentially further the field by establishing MBCT as a treatment option for this group and shift focus towards implementation research.

o “*Minor adaptations*” of MBCT for current episodes need operational criteria for reproducibility (what qualifies as “minor”? who adjudicates? thresholds for exclusion). Without this, between-study heterogeneity risks becoming uninterpretable.

Please note that “minor adaptations” explicitly refers to *delivery-related* adjustments (e.g. rephrasing content to address current symptoms rather than relapse, providing practice support appropriate to the current state of the patient). We have italicised the term “delivery” in the relevant section of the protocol (p. 11, para. 2).

The protocol explicitly states that studies that alter structure or content of the manual will be excluded, which is consistent with prior meta-analyses of MBCT in current depression (e.g. Goldberg et al., 2019, see also reviews by Kuyken et al., 2016, and McCartney et al., 2021). We have italicised the phrase “without altering the structure or core content of the manual”.

Adjudication is handled within the study selection process. As described in the “Study selection” section, studies were screened by two independent reviewers, with disagreements about eligibility being referred to a third reviewer whose decision was final.

2. *Critical: There are major discrepancies between the PROSPERO registration and the current protocol:*

o *Population: The introduction of the DTD construct represents a substantial shift from the original registration, which did not specify this population.*

As described further above, the original PROSPERO registration in 2022 used the terms “chronic and treatment-resistant depression”, capturing the wider spectrum of persistent and treatment-resistant courses. Since then, the field has moved to adopt the concept of DTD, which represents a wider heuristic and is not linked to an objective way of assessment. Our operationalisation continues to use conventional definitions of chronic depression and the broader range of definitions of treatment resistance, including treatment non-response and non-remission (see McIntyre et al., 2023). As stated further above, the move to DTD is a terminological update, not a change in population.

The PROSPERO registration has been updated for consistency and the title now uses DTD: “An independent patient data (IPD) meta-analysis of Cognitive Behavior Analysis of Psychotherapy

(CBASP) and Mindfulness-Based Cognitive Therapy (MBCT) for patients with difficult-to-treat depression (DTD)” (<https://www.crd.york.ac.uk/PROSPERO/view/CRD42022332039>)

o Primary outcomes: The selective reporting of outcomes—particularly the omission of deterioration, suicidality, and adverse events, which are relegated to a separate protocol—constitutes outcome switching.

The protocol clearly states that the PROSPERO registration represents a wider project. It is usual practice for such projects to address different objectives in separate review papers. The two separate protocols were both submitted to BMJOpen on 10 June 2025.

In the “Outcomes” section (p. 12, para. 2), the current protocol clearly states that “we will derive dichotomous outcomes reflecting remission, improvement and *deterioration*”. By including this information on negative effects, the protocol goes beyond common practice in existing meta-analyses of MBCT. However, detailed analyses of different aspects of negative aspects are beyond the scope of this protocol and are covered in detail in a separate protocol. This means that the project is giving increased emphasis to this often-overlooked aspect, rather than “switching” outcomes.

All outcomes are prospectively documented (see the “Main outcomes section of the PROSPERO registration), cross-referenced, and reported. We have added a clarifying sentence to the protocol (p. 12, para. 3):

“A more detailed analysis of negative effects including suicidality is covered in a separate protocol covering a network meta-analysis of MBCT and another individual psychotherapeutic intervention (Michalak et al., in submission).”

o Interventions and comparators: MBCT is repositioned from comparator to primary intervention, while CBASP, originally central in PROSPERO, is omitted here.

Thank you for pointing this out. The PROSPERO record has been updated: MBCT is listed as the intervention and CBASP and other interventions as comparator(s) (see <https://www.crd.york.ac.uk/PROSPERO/view/CRD42022332039>). This inconsistency may have occurred because objectives of the registration not covered in this protocol will be investigated using network meta-analysis, where interventions are conceptualised in terms of nodes instead of a strict designation of intervention and comparator.

o Moderators: Additional moderators not prespecified in PROSPERO are introduced, raising concerns of post hoc rationalization.

The PROSPERO entry listed moderators as examples, as indicated by the phrase “such as”, and was not intended as exhaustive. The current protocol provides the complete list, along with citations of the evidence that justifies each inclusion. This is transparently stated (p. 10, para. 1):

“The moderator and sensitivity analyses described here were prespecified in this protocol prior to data analysis but were not detailed in the initial PROSPERO entry.”

o Allegiance and COI: Multiple investigators have deep MBCT involvement. The plan to extract trial-level funding/COI is welcome but must include a priori sensitivity analyses (e.g., excluding high-allegiance studies; meta-regression on allegiance).

We agree on the importance of transparency and will extract and report trial-level COI and funding information as described. However, allegiance is not identified in current reporting guidelines for systematic reviews or IPD meta-analyses (e.g., PRISMA, PRISMA-IPD, Cochrane Handbook, APA MARS) as a required domain for risk of bias or sensitivity analysis. As allegiance is closely entangled with expertise and treatment fidelity, we do not believe pre-specifying allegiance-based sensitivity analyses is appropriate. Instead, we will ensure comprehensive reporting of potential COI so that readers can interpret results with this context in mind.

Although the authors frame these as “refinements” within a broader project, splitting outcomes and interventions across multiple protocols risks diluting the integrity of prospective registration. The protocol, in its current form, may no longer function as a prospective safeguard against bias.

The PROSPERO registration documents all objectives and the separate protocol manuscripts do not add new objectives post hoc; rather, they provide space to describe individual analyses in sufficient methodological detail for transparency and reproducibility. This practice is consistent with common approaches in complex review programmes (including IPD consortia and living systematic reviews), where a single overarching registration supports multiple sub-protocols or linked papers. We believe this structure strengthens, rather than weakens, the prospective safeguard against bias, as each analytic component is fully specified, peer-reviewed, and published rather than remaining subsumed in a single, broad protocol, that would need to remain vague or risk becoming unwieldy.

Importantly, BMJ Open guidelines (2025) state that “Protocol papers should report planned or ongoing studies. The dates of the study should be included in the manuscript.” The retrospective registration after searches had already been conducted, combined with the deviations outlined above, is inconsistent with this principle.

We respectfully clarify that registration was completed on 21 May 2022, prior to the first formal search (17 June 2024). The protocol was posted to medRxiv on 9 June and submitted to BMJ Open on 10 June 2025 (dates reported in the manuscript). Given this chronology, the registration predates the initiation of formal searches; we therefore do not agree that our registration is retrospective in relation to the searches.

Consistent with BMJ Open practice for IPD meta-analyses, we disclosed that a subset of IPD had been received prior to protocol publication and submission. Publishing IPD meta-analysis protocols after receipt of IPD in BMJ Open is well established (e.g., Breedvelt et al., 2020, *BMJ Open*2020;10:e034158). We chose the journal based on the understanding that the journal’s protocol guidance emphasises transparent reporting of study dates rather than prohibiting ongoing data receipt

in prospectively registered programmes. We have described the noted procedural “deviations” in detail and confirm they do not reflect a change in scope or any lack of transparency; any amendments are documented and time-stamped in the registry record.

Recommendation: Major revision. The protocol and the PROSPERO entry must be fully aligned—either by updating PROSPERO to reflect the actual scope, population, outcomes, and analyses of this protocol, or by revising the manuscript to correspond faithfully to the original registration.

As described above, the protocol and the PROSPERO entry (<https://www.crd.york.ac.uk/PROSPERO/view/CRD42022332039>) are now fully aligned.

Separate protocols for different interventions or outcomes should not be used as a workaround.

Splitting methods into linked protocols is not a workaround but a pragmatic approach for complex programmes. Our PROSPERO record prospectively lists all objectives (with timestamped amendments), and the separate manuscripts do not add aims post hoc. Each protocol pre-specifies a discrete analysis with full methods, cross-references the umbrella registration, which will contain links to sister protocols, and commits to reporting all planned analyses irrespective of results. This structure strengthens, rather than weakens, the prospective safeguard against bias by replacing a single, vague plan with two fully specified plans, that we have subjected to peer-review by submitting both to BMJOpen.

Furthermore, the manuscript should clearly articulate the research gap in the treatment of DTD, acknowledging the weaknesses of current evidence and specifying how this review will address them.

The research gap and weakness of the current evidence is clearly articulated: there is considerable evidence on the use of MBCT in DTD but no systematic review of this evidence - multiple RCTs of MBCT in DTD exist but have not been coherently synthesised, and trial-level heterogeneity (definitions, outcomes, comparators, follow-up) and underpowered moderator analyses limit inferences. Our introduction highlights these points and makes it clear that the synthesised evidence, if positive, is likely to be sufficient to justify guideline endorsement of MBCT as a further-line treatment for depression, meaning that the review has a strong potential for impact.

We have strengthened the relevant paragraphs as follows (p. 8, para. 2):

“Based on this reasoning, research has extended the use of MBCT to patients with current depression described as ‘difficult-to-treat’. Preliminary studies investigating MBCT in treatment-resistant and chronic depression were published in 2008 [17] and 2009 [18] and several definitive randomised controlled trials have been conducted since then [19–24], suggesting that evidence has accumulated to a point where the use of MBCT as a treatment for patients with current DTD can now be considered for guideline endorsement and wider implementation. **However, conclusions from individual studies are constrained by heterogeneous definitions and settings, variable comparators, inconsistent outcome reporting, and limited power for moderator analyses.** It seems timely therefore to bring data from existing randomised controlled trials together and analyse outcome across studies. For this purpose, we have pooled individual patient data (IPD) from these studies to conduct meta-analyses with one-stage random effects meta-analysis of independent patient data (IPD) as our

primary approach given its advantages over conventional aggregate data meta-analyses [25–28]. **This IPD meta-analysis will apply a consistent DTD framework, harmonise outcomes and test prespecified patient-level moderators, with comprehensive risk-of-bias and small-study assessments.** The main aim of the analyses is to establish efficacy of MBCT for patients with DTD as compared to the control conditions utilised in the existing trials. Given the need for both effective symptom reduction and the maintenance of gains, we will aim to determine immediate effects at the end of the eight-week MBCT intervention (post-treatment) and effects in the longer-term after patients have finished the intervention and are free to continue engaging in mindfulness practice by themselves (follow-up).”

Full alignment and a stronger rationale are necessary to preserve the integrity and purpose of prospective registration.

We have fully aligned the PROSPERO registration (<https://www.crd.york.ac.uk/PROSPERO/view/CRD42022332039>) and the protocol.

VERSION 3 - REVIEW

Reviewer	1
Name	Villalón López, Francisco
Affiliation	Diego Portales University
Date	23-Oct-2025
COI	

Efficacy and Moderators of Mindfulness-Based Cognitive Therapy (MBCT) in ‘Difficult-to-Treat’ Depression: Protocol for a Systematic Review and Individual Patient Data Meta-analysis of Randomised Controlled Trials” Registered PROSPERO: CRD42022332039

Thank you to the research team for their excellent work on this important and timely topic. I carefully reviewed the updated protocol and am satisfied with most of the revisions compared with the previous version. The current version clearly addresses most concerns regarding the research update, clarification of the research gap and the DTD definition, and minor methodological adaptations.

However, the conceptual pivot from *chronic and treatment-resistant depression* to *difficult-to-treat depression (DTD)* could be confusing. Readers might interpret that MBCT is effective for all

forms of DTD rather than for the specific definition used in this study. The title or framing might thus convey a broader concept than intended — a kind of “Trojan horse,” where a complex idea is introduced that could obscure the main message. A clearer and more straightforward conceptual framing would improve scientific communication, although I acknowledge that this is ultimately a decision for the authors and editors, while readers should interpret the findings cautiously. This is also recommended in Cochrane Guidelines. The shift toward DTD is understandable and relevant to current discourse, but it might be more appropriate to highlight this transition in the *Discussion* rather than in the title. That said, this is not a major issue for publication but rather a difference in viewpoint.

Regarding the therapeutic alliance, I agree that it is an important consideration, but its inclusion is at the authors’ discretion. Readers should interpret this aspect with caution. For instance, in a previous meta-analysis of CBT with ERP for obsessive–compulsive disorder, allegiance showed a strong effect ($g = 0.95$), whereas non-allegiance had virtually no effect ($g = 0.02$; Reid et al., 2021).

Concerning the discrepancy between the PROSPERO registration and the current protocol: while such practices occasionally occur, methodological guidelines recommend defining a **specific PICO question** for each systematic review. If two distinct protocols are derived, they should be **prospectively and a priori divided**, referencing the original registration and explaining the differences.

Currently, in your PICO question, the *Intervention* is defined as MBCT and the *Comparator* as CBASP and others. However, this does not align with the search strategy, since CBASP was not included as a comparator in the MBCT search, and in a future CBASP protocol, CBASP would be defined as an intervention, not as a comparator.

The response to the reviewer stated:

“This inconsistency may have occurred because objectives of the registration not covered in this protocol will be investigated using network meta-analysis, where interventions are conceptualised in terms of nodes instead of a strict designation of intervention and comparators.”

If the intent is to conduct a **network meta-analysis** comparing MBCT and CBASP as interventions against any control condition (e.g., psychological placebo), it would be better to state this explicitly. Moreover, network meta-analyses should be clearly identified **a priori** in the PROSPERO registration, as their scope differs from pairwise meta-analysis. In such cases, the PICO question should specify “*any intervention*” rather than a fixed pair.

The current PROSPERO record contains major changes and encompasses more than one research question, which is methodologically problematic. The **Cochrane Handbook for Systematic Reviews of Interventions (Chapter 2 and 4)** recommends registering one clearly defined PICO question per protocol and provides guidance on “lumping” and “splitting” strategies or splitting and merging.

Similarly, PROSPERO (Booth et al., 2012) defines a new review as one involving:

“Addition of new treatment comparisons, substantial changes to the population, changes in inclusion/exclusion criteria, or introduction of new analytic techniques such as switching from aggregate to individual-participant data meta-analyses.”

Therefore, this remains a critical issue. I strongly recommend updating the PROSPERO registration to ensure full alignment — either by (1) revising the existing record, (2) creating separate protocols for distinct PICO questions (e.g., MBCT alone, CBASP alone, or both), or (3) preparing a comprehensive protocol encompassing all pre-registered interventions and outcomes. If a new synthesis method such as network meta-analysis is planned, a separate pre-registered protocol should also be created.

You can cite the previous PROSPERO record for transparency.

If you wish to maintain the current protocol, I suggest specifying the PICO elements explicitly in the PROSPERO registration as follows:

1. Protocol 1:

P: DTD

I: MBCT

C: Any comparator

O: As already defined.

For other protocols:

2. Protocol 2:

P: DTD

I: Cognitive Behavioral Analysis System of Psychotherapy (CBASP)

C: Any comparator

O: As already defined.

3. Protocol 3 (combined interventions):

P: DTD

I: CBASP or MBCT

C: Any comparator

O: As already defined.

For each protocol, please indicate whether a **network meta-analysis** is preplanned as part of the synthesis strategy and heterogeneity assessment. Including this explicitly in the registration will improve transparency and alignment with methodological guidelines.

VERSION 3 - AUTHOR RESPONSE

Comments to the Author:

Efficacy and Moderators of Mindfulness-Based Cognitive Therapy (MBCT) in ‘Difficult-to-Treat’ Depression: Protocol for a Systematic Review and Individual Patient Data Meta-analysis of Randomised Controlled Trials” Registered PROSPERO: CRD42022332039

Thank you to the research team for their excellent work on this important and timely topic. I carefully reviewed the updated protocol and am satisfied with most of the revisions compared with the previous version. The current version clearly addresses most concerns regarding the research update, clarification of the research gap and the DTD definition, and minor methodological adaptations.

Thank you for your careful review. We are pleased that the revisions addressed the key concerns regarding the rationale, DTD definition, and methodological refinements, and we appreciate your constructive engagement.

However, the conceptual pivot from chronic and treatment-resistant depression to difficult-to-treat depression (DTD) could be confusing. Readers might interpret that MBCT is effective for all forms of

DTD rather than for the specific definition used in this study. The title or framing might thus convey a broader concept than intended — a kind of “Trojan horse,” where a complex idea is introduced that could obscure the main message. A clearer and more straightforward conceptual framing would improve scientific communication, although I acknowledge that this is ultimately a decision for the authors and editors, while readers should interpret the findings cautiously. This is also recommended in Cochrane Guidelines. The shift toward DTD is understandable and relevant to current discourse, but it might be more appropriate to highlight this transition in the Discussion rather than in the title. That said, this is not a major issue for publication but rather a difference in viewpoint.

Thank you for highlighting the potential for conceptual ambiguity and for acknowledging that this reflects a difference in viewpoint rather than a substantive barrier to publication.

As previously discussed, we will ensure clear framing to support accurate interpretation in the final manuscript by explicitly linking DTD to the conventional definitions used in the included studies. The IPD design also allows empirical examination of whether definitional subtypes (e.g., chronicity, non-response) moderate treatment effects; this moderator is prespecified.

Regarding the therapeutic alliance, I agree that it is an important consideration, but its inclusion is at the authors’ discretion. Readers should interpret this aspect with caution. For instance, in a previous meta-analysis of CBT with ERP for obsessive–compulsive disorder, allegiance showed a strong effect ($g = 0.95$), whereas non-allegiance had virtually no effect ($g = 0.02$; Reid et al., 2021).

We thank the reviewer for highlighting the relevance of therapist factors, including therapeutic alliance and treatment allegiance, and for drawing attention to the findings from Reid et al. (2021). We agree that allegiance effects are an important consideration in psychological treatment research.

We will note this issue in the narrative synthesis and will report trial funding and conflicts of interest transparently in the final manuscript as previously suggested.

Concerning the discrepancy between the PROSPERO registration and the current protocol: while such practices occasionally occur, methodological guidelines recommend defining a specific PICO question for each systematic review. If two distinct protocols are derived, they should be prospectively and a priori divided, referencing the original registration and explaining the differences.

Our PROSPERO registration specifies three objectives: objectives 1 and 2 concern negative outcomes (deterioration and suicidality) and objective 3 concerns depressive symptom change. The two linked protocols correspond directly to these pre-registered objectives: one addresses negative outcomes using network meta-analysis, and the other addresses symptom change using pairwise comparisons (from the MBCT perspective, the CBASP perspective will be added).

We agree that it might have been clearer to highlight explicitly in the initial PROSPERO record that separate protocols would be used for these objectives. Nevertheless, it is not unusual that greater specificity about analytic structure emerges during protocol development, provided the review objectives remain unchanged. In this case, the division reflects emphasis and methodological requirements rather than a post-hoc change in review aims (see Cochrane Handbook excerpt below).

We appreciate the reviewer’s suggestion to clarify this in the documentation and have ensured that the linkage between the registration and protocols is made more explicit (see responses further below).

“There may be a need to modify the comparisons and even add new ones at the review stage in light of the data that are collected. For example, important variations in the intervention may be discovered only after data are collected, or modifying the comparison may facilitate the possibility of synthesis when only one or few studies meet the comparison PICO. Planning for the latter scenario at the protocol stage may lead to less post-hoc decision making (Chapter 2, Section 2.5.3) and, of course, any changes made during the conduct of the review should be recorded and documented in the final report.” https://www.cochrane.org/authors/handbooks-and-manuals/handbook/current/chapter-03?utm_source=chatgpt.com#section-3-1

Currently, in your PICO question, the Intervention is defined as MBCT and the Comparator as CBASP and others. However, this does not align with the search strategy, since CBASP was not included as a comparator in the MBCT search, and in a future CBASP protocol, CBASP would be defined as an intervention, not as a comparator.

The response to the reviewer stated:

“This inconsistency may have occurred because objectives of the registration not covered in this protocol will be investigated using network meta-analysis, where interventions are conceptualised in terms of nodes instead of a strict designation of intervention and comparators.”

If the intent is to conduct a network meta-analysis comparing MBCT and CBASP as interventions against any control condition (e.g., psychological placebo), it would be better to state this explicitly. Moreover, network meta-analyses should be clearly identified a priori in the PROSPERO registration, as their scope differs from pairwise meta-analysis. In such cases, the PICO question should specify “any intervention” rather than a fixed pair.

It is important to clarify that the intention is to investigate negative effects (objectives 1 and 2 of the PROSPERO registration) using a network meta-analysis as primary approach (described in a separate protocol linked to the registration), while the current protocol (linked to objective 3) will use pairwise comparison to investigate the effectiveness of MBCT. This approach reflects the high clinical and policy relevance of MBCT for DTD. A dedicated pairwise comparison allows us to address a clinically critical treatment decision, to examine effects without relying on network connectivity and transitivity assumptions, and to conduct pre-specified moderator analyses that are not feasible within a network structure. For clarification, we have added the following sentence to the objectives section of the PROSPERO registration:

“Objectives 1 and 2 will be addressed using network meta-analysis as the primary approach, while objective 3 will be addressed using pairwise comparisons as the primary approach (see the separate protocol linked to this registration).”

We agree with the reviewer regarding the need to clearly distinguish analytic approaches and their associated PICOs. In line with this advice, we have revised the eligibility (PICO) section of the PROSPERO registration to map the analytic strategy for each registered objective more explicitly (for more details see our response to the related point below).

The current PROSPERO record contains major changes and encompasses more than one research question, which is methodologically problematic. The Cochrane Handbook for Systematic Reviews of Interventions (Chapter 2 and 4) recommends registering one clearly defined PICO question per protocol and provides guidance on “lumping” and “splitting” strategies or splitting and merging.

Similarly, PROSPERO (Booth et al., 2012) defines a new review as one involving: “Addition of new treatment comparisons, substantial changes to the population, changes in inclusion/exclusion criteria, or introduction of new analytic techniques such as switching from aggregate to individual-participant data meta-analyses.” Therefore, this remains a critical issue.

The examples cited in Booth et al. (2012) involve substantial changes in review scope, such as moving from aggregate to individual participant data. In our case, the review objectives remain unchanged, and the analytic refinement concerns methodology rather than review scope. We have therefore followed the reviewer’s suggestion further below to explicitly distinguish PICO instead of registering a new review.

I strongly recommend updating the PROSPERO registration to ensure full alignment — either by (1) revising the existing record, (2) creating separate protocols for distinct PICO questions (e.g., MBCT alone, CBASP alone, or both), or (3) preparing a comprehensive protocol encompassing all pre-registered interventions and outcomes. If a new synthesis method such as network meta-analysis is planned, a separate pre-registered protocol should also be created.

You can cite the previous PROSPERO record for transparency. If you wish to maintain the current protocol, I suggest specifying the PICO elements explicitly in the PROSPERO registration as follows:

1. Protocol 1:

P: DTD

I: MBCT

C: Any comparator

O: As already defined.

For other protocols:

2. Protocol 2:

P: DTD

I: Cognitive Behavioral Analysis System of Psychotherapy (CBASP)

C: Any comparator

O: As already defined.

3. Protocol 3 (combined interventions):

P: DTD

I: CBASP or MBCT

C: Any comparator

O: As already defined.

For each protocol, please indicate whether a network meta-analysis is preplanned as part of the synthesis strategy and heterogeneity assessment. Including this explicitly in the registration will improve transparency and alignment with methodological guidelines.

We agree that the PROSPERO eligibility description required updating to reflect both network and pairwise structures. Unfortunately, the headings of the eligibility section are fixed and do not accommodate multiple full PICO schemas. We have therefore implemented the request in the following format:

“Interventions/exposures

Mindfulness-Based Cognitive Therapy (MBCT) and Cognitive Behavioral Analysis System of Psychotherapy (CBASP) are the primary interventions of interest. Other structured psychological interventions will be included when they contribute to the network for safety outcomes.

Comparators/control

For the network meta-analysis, all eligible interventions will form nodes in a connected network. For the pairwise analyses testing effectiveness of MBCT, only studies with direct MBCT comparisons will be included. For the pairwise analyses testing effectiveness of CBASP, only studies with direct CBASP comparisons will be included.”

“Main outcomes

[...] Deterioration, suicidality, adverse events will be analysed with network meta-analysis as a primary approach; depressive symptom severity will be analysed with pairwise IPD meta-analysis.”

We trust that the revised manuscript now meets all requirements for publication and thank the editor and reviewer again for their thoughtful and constructive feedback throughout this process.